# CASPO : CONFIDENCE-AWARE STEP-WISE PREFERENCE OPTIMIZATION FOR RELIABLE REASONING IN LARGE LANGUAGE MODELS

## ABSTRACT

Large language models (LLMs) have demonstrated strong performance on tasks requiring multi-step reasoning, from mathematical derivations to knowledge-intensive open-domain generation. However, even when LLMs produce correct final answers, their reasoning processes often involve uncertain or inconsistent steps, which makes them prone to failure when facing similar problems again. To address this issue, we introduce CASPO , a framework that incorporates step-wise confidence into both training and inference. During training, CASPO constructs confidence-filtered preference pairs that capture both correct but low-confidence predictions and incorrect predictions, and optimizes them through iterative Direct Preference Optimization. During inference, we propose Confidence-aware Thought (CaT) strategy that prunes low-confidence reasoning trajectories to enhance reliability. Experiments on 10 reasoning benchmarks and across diverse model families show that CASPO yields improvements in both step-wise faithfulness and final-answer accuracy. We also release a step-wise dataset with confidence annotations to facilitate fine-grained analysis of model reasoning and expose hidden inconsistencies in existing benchmarks.

## 1 INTRODUCTION

LLMs such as OpenAI-O1 (Jaech et al., 2024), DeepSeek-R1 (Guo et al., 2025), and Kimi-K1.5 (Team et al., 2025) have achieved remarkable progress on complex reasoning tasks, ranging from commonsense inference and mathematical problem solving to multi-hop question answering (Liu et al., 2025; Xiong et al., 2025; Hu et al., 2025). However, prior studies (Arcuschin et al.) reveal that correct final answers often conceal flawed intermediate reasoning. LLMs may mix valid and invalid derivations, yielding reasoning traces that are inconsistent, fabricated, or only loosely connected to the solution. This discrepancy highlights a critical risk: even if models occasionally reach the right conclusion, their uncertainty and misalignment at the step wise mean they may easily fail when faced with similar problems again. In high-stakes domains such as medical diagnosis, finance, and scientific research, such unreliability undermines trust, interpretability, and safe deployment (Blundell et al., 2015; Tomani & Buettner, 2021; Fadeeva et al., 2024).

Recent advances have primarily enhanced reasoning reliability at the trajectory level, treating the reasoning process as a whole. Chain-of-thought prompting (CoT) (Wei et al., 2022) improves performance by eliciting explicit intermediate steps, while self-consistency (Wang et al., 2022) stabilizes accuracy by aggregating multiple reasoning paths via majority voting. Preference-based optimization methods such as Group Relative Policy Optimization (GRPO) (Guo et al., 2025) further align outputs with preferred reasoning trajectories using verifiable rewards. More sophisticated frameworks have also emerged to optimize full reasoning paths, rStar-Math (Guan et al., 2025) introduces self-evolution through deep thinking. While this trajectory-centric paradigm overlooks the fine-grained reliability of individual reasoning steps, which are essential for ensuring logical soundness and generalizability across tasks. Consequently, answers may still be rest on or be distorted by partially flawed derivations, yielding solutions that appear correct but are logically unsound.

To address this limitation, several studies have introduced step-wise supervision to enhance intermediate reasoning quality. Step-wise preference optimization methods (Razghandi et al., 2025)

provide valuable intermediate guidance, while process-based self-rewarding frameworks incorporate step-wise evaluation into reinforcement learning pipelines (Tu et al., 2025). Weakness-driven augmentation strategies such as SwS (Liang et al., 2025) also highlight the benefits of diagnosing and addressing systematic reasoning failures. Nevertheless, these methods remain limited: most depend on heuristic feedback and do not explicitly connect step correctness with confidence, which often results in confident but inconsistent reasoning.

Parallel efforts (Xu et al., 2024) have explored confidence estimation via token probabilities, but empirical evidence (Arcuschin et al.; Li et al., 2025a) and our analysis show that token-level confidence often reflects surface fluency or frequent patterns rather than genuine reasoning reliability. LLMs often assign high probabilities to superficial or frequent tokens that are syntactically well-formed but semantically irrelevant to the reasoning process, while underestimating uncertainty in steps that are crucial for maintaining consistent and valid reasoning. This miscalibration highlights the need for a principled approach to step-wise confidence modeling.

In light of these findings, our main insight is to enhance step-wise certainty by explicitly aligning model confidence with step correctness, and by leveraging this alignment to guide reasoning during inference. Motivated by this idea, we propose CASPO (Confidence-Aware Step-wise Preference Optimization), a framework that integrates step-wise confidence into both training and inference.

During training, CASPO aligns confidence with correctness by constructing preference pairs that include both correct-but-uncertain and incorrect predictions, which are optimized via iterative Direct Preference Optimization (DPO). During inference, we introduce a Confidence-aware Thought (CaT) strategy that uses cumulative confidence to guide the expansion and pruning of reasoning trajectories. Together, this two-stage design directly addresses the tension between exploration and reliability, ensuring that step-wise improvements propagate into more faithful and robust final answers.

Our contributions are summarized as follows: First, we propose CASPO, a framework that explicitly incorporates step-wise confidence into both training and inference. By aligning confidence with correctness during training and leveraging cumulative confidence to guide path expansion and pruning during inference, the method effectively mitigates the tension between exploration and reliability. Second, experiments on ten reasoning benchmarks show consistent gains: e.g., on Qwen2.5-7B-Instruct, CASPO improves average accuracy from 44.4 to 50.6 (+6.2), and further reaches 56.1 with confidence-aware inference. Moreover, our method demonstrates robust improvements not only on out-of-domain tasks but also across different model families, indicating its generality and adaptability. Finally, we release a step-wise dataset annotated with confidence scores, which provides fine-grained supervision and reveals hidden inconsistencies in existing reasoning benchmarks. Code is available at : https://anonymous.4open.science/r/CASPO-1O2K.

## 2 RELATED WORK

**Large Reasoning Models.** The development of large reasoning models (LRMs) has progressed from simple prompting to more sophisticated strategies. CoT showed that explicit step-by-step reasoning improves performance on complex tasks, while Self-Consistency (Wang et al., 2022) enhanced robustness by aggregating multiple reasoning paths. Recent systems such as OpenAI's o1 (Jaech et al., 2024) and DeepSeek-r1 (Guo et al., 2025) emphasize post-training methods that elicit extended reasoning traces for greater transparency and accuracy. In parallel, distillation techniques (Hsieh et al., 2023) transfer high-quality reasoning trajectories to smaller models for efficiency. For instance, (Guan et al., 2025) explicitly utilizes rationales from large teacher models to supervise smaller students, reducing data requirements while maintaining performance. Structured approaches such as Tree-of-Thoughts (Yao et al., 2023), Graph-of-Thoughts (Besta et al., 2024), and reinforcement learning (Zhang et al., 2024; Li et al., 2025b; Zhang et al., 2025) further expand the reasoning space, albeit often at high computational cost.

**Reasoning Process Verification.** As reasoning traces grow longer, ensuring their faithfulness and reliability has become increasingly important. One line of work introduces process reward models (PRMs) (Lightman et al., 2023; Wang et al., 2023), trained on datasets such as PRM800K (Lightman et al., 2023), to score intermediate reasoning steps; works like PURE (Cheng et al., 2025) refine step-wise credit assignment in reinforcement learning. Complementing these scoring methods, approaches such as (Patnaik et al., 2025a;b) demonstrate that models can significantly enhance

reliability by learning from collaborative deliberation or selective rationale optimization, effectively utilizing mutual verification to distill high-quality reasoning patterns through preference ranking. Another direction explores self-verification (Lightman et al., 2023; Qu et al., 2024), where models attempt to detect and correct their own errors through reflection (Du et al., 2023) and consistency checks (Saunders et al., 2022). While fully autonomous self-correction remains challenging, recent studies suggest that combining self-verification with lightweight external supervision can improve reasoning reliability without incurring the heavy cost of large reward models.

**Verification-Enhanced Reasoning.** Beyond evaluation, recent work integrates verification directly into reasoning. Test-time scaling generates multiple candidate solutions and selects the most reliable one, improving accuracy at high computational cost. At training time, reinforcement learning with verifiable rewards (e.g., SimpleRL (Zeng et al., 2025), PURE (Cheng et al., 2025)) has been shown to iteratively refine reasoning ability by filtering or rewarding faithful traces. To further reduce reliance on explicit reward models, DPO-based approaches approximate reward signals with likelihood estimation, lowering overhead. Co-training frameworks that jointly optimize generator and verifier have also been explored (Ouyang et al., 2022), although challenges of scalability and stability remain. Grounded in these directions, CASPO differs from these collaborative or external-distillation approaches: rather than relying on multi-model collaboration or mimicking teacher preferences, it unifies training and inference through *intrinsic step-wise confidence calibration*, utilizing the student's own token entropy to guide reliable reasoning paths.

## 3 METHODS

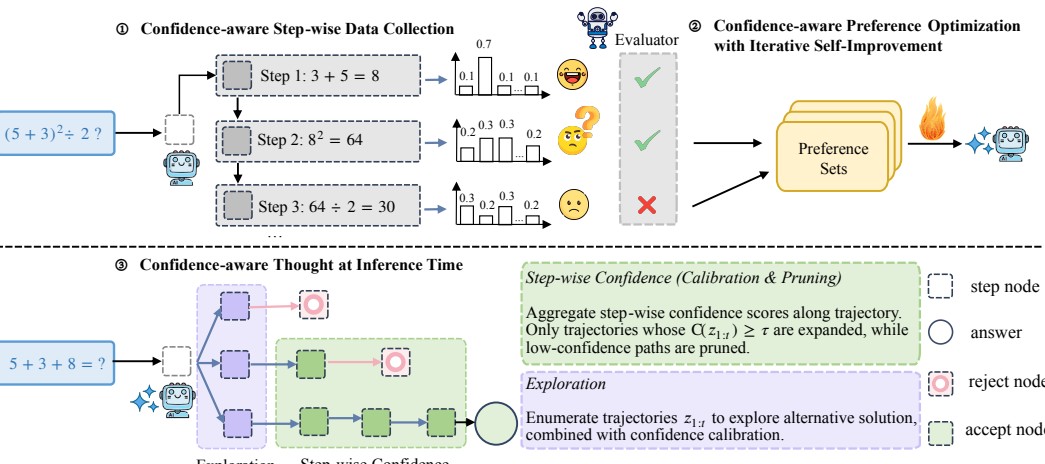

Figure 1: Overview of CASPO : Confidence-Guided Reasoning and Preference Optimization

In this section, we present the core design of CASPO , which integrates confidence estimation into both training and inference. As illustrated in Figure 1, our framework consists of two main phases: (i) Training with confidence-aware preference optimization, where step-wise data collection captures both low-confidence correctness and confident mistakes to guide preference learning, and (II) inference with a confidence-aware thoughts (CaT) strategy, where reasoning paths are dynamically expanded or pruned based on cumulative confidence.

### 3.1 MOTIVATION AND PRE-ANALYSIS

Recent progress (Wang et al., 2022; Zuo et al., 2025; Li et al., 2025b) in LRMs have demonstrated that sampling multiple candidate chains of thought can boost both robustness and accuracy. However, this practice also makes a central tension explicit: how to balance broad exploration of reasoning with faithful, stepwise verification. Excessive diversity invites plausible yet flawed traces, whereas excessive consistency collapses search into overly narrow trajectories. Either failure mode undermines reliability in real-world settings.

To address this issue, our objective is to enable models to assign a calibrated confidence estimate to each step within the reasoning process. Importantly, the framework seeks to ensure that high-confidence predictions are more reliably associated with correct reasoning. We propose a paradigm that integrates structured method enumeration with fine-grained, step-wise confidence estimation and external verification. The goal is twofold: to preserve the breadth of exploration across possible solution routes, while more accurately identifying and retaining genuinely correct reasoning. This formulation provides the foundation for the method described in the next section.

### 3.2 CASPO : CONFIDENCE-AWARE STEP-WISE PREFERENCE OPTIMIZATION

**Confidence-aware Step-wise Data Collection.** We consider an auto-regressive language model $\theta$, which defines a next-token distribution $\pi_\theta(\cdot|x)$ given an input prompt $x$. Then, for each query $x \in \mathcal{D}$, suppose the reasoning path contains $m$ steps. At step $i \in [1, m]$, the model conditions on the original problem and the partial chain-of-thought generated so far, forming an intermediate $q_i$, and produces a corresponding step-wise answer $s_j$, where $j \in [1, m]$. These step-wise predictions collectively lead to the final answer $a$. During generation, let the step answer $s_j$ consist of tokens $\{t_l\}_{l=1}^L$, the confidence of this answer $s_j$ is computed by the token entropy:

$$\text{confidence}(s_j|q_j) = -\frac{1}{L} \sum_{l=1}^L \sum_{v \in \mathcal{V}} \pi_\theta(v|q_j) \log \pi_\theta(v|q_j) \tag{1}$$

Where $L$ is the length of the step answer, $\mathcal{V}$ is the vocabulary, and $\pi_\theta(v|q_j)$ denotes the predictive distribution over tokens $v$. Higher cumulative entropy indicates greater uncertainty and, hence, lower confidence in the generation. We adopt token-level entropy as our uncertainty metric because it captures the model's intrinsic uncertainty during generation, avoiding the overconfidence bias and hallucination sensitivity inherent in frequency-based diversity measures. This reference-free criterion evaluates each candidate's confidence independently of the ground truth.

To obtain reliable supervision, we employ Qwen2.5-Math-7B-Instruct as an external evaluator. The evaluator verifies whether the step-wise answer $s_i$ is correct, and the model $\theta$ gives the confidence to the corresponding question $q_j$:

- If $s_j$ if correct and high confidence, omit.
- If $s_j$ if correct but has low confidence, set $y_w$ to $s_j$, and $y_l$ to the second highest probability word in $\pi_\theta(\cdot|x)$.
- If $s_j$ if incorrect, set $y_w$ to the correct answer and $y_l$ to $s_j$.

**Preference Optimization with Iterative Self-Improvement.** Based on the step-centric dataset $\mathcal{D}$ constructed in Algorithm 1, we form preference pairs $(q_j, y_j^w, y_j^l)$. This design ensures that both reliable but uncertain predictions and confidently wrong predictions contribute to preference learning.

The training objective follows the DPO formulation, which encourages the target model $\pi_\theta$ to increase the relative likelihood of the preferred answer compared to the dispreferred one:

$$\mathcal{L}_{\text{DPO}} = -\log \sigma \left( \beta \left[ \log \frac{\pi_\theta(y_j^w|q_j)}{\pi_{\text{ref}}(y_j^w|q_j)} - \log \frac{\pi_\theta(y_j^l|q_j)}{\pi_{\text{ref}}(y_j^l|q_j)} \right] \right), \tag{2}$$

where $\beta$ controls the strength of preference alignment.

To enable iterative self-improvement, we adopt an *Iterative DPO* scheme: at each iteration $k$, the target model $\pi_{\theta_k}$ is optimized using the above loss with respect to the previous model $\pi_{\text{ref}} = \pi_{\theta_{k-1}}$ as the reference. After optimization, we set $\pi_{\text{ref}} \leftarrow \pi_{\theta_k}$ for the next step. This iterative update makes the reference distribution progressively stronger, allowing the model to bootstrap its own improvements and continuously refine its reasoning preferences.

**Confidence-aware Thought.** After iterative preference optimization, the model not only learns to prefer correct reasoning steps but also calibrates its confidence estimation at each step. This enables a *Confidence-Aware Thought* (CaT) inference strategy: instead of committing to a single linear chain,

the model explores a reasoning tree where each node corresponds to a partial reasoning trajectory $z_{1:t} = (z_1, \ldots, z_t)$ with an associated confidence score

$$C(z_{1:t}) = \prod_{i=1}^{t} \text{confidence}(z_i | z_{1:i-1}), \qquad (3)$$

where $\text{conf}(z_i | z_{1:i-1})$ denotes the normalized confidence of reasoning step $z_i$ given the previous context.

During inference, a path is expanded only if its cumulative confidence $C(z_{1:t})$ exceeds a threshold $\tau$:

$$C(z_{1:t}) \geq \tau, \qquad (4)$$

otherwise the path is pruned. In this way, the model self-regulates its reasoning: high-confidence trajectories are pursued, while uncertain or misleading trajectories are terminated early.

This confidence-aware pruning naturally acts as a *self-improvement* mechanism. Since paths with insufficient confidence are discarded, the search process prioritizes reliable reasoning steps without requiring external supervision. The resulting reasoning tree thus balances exploration and reliability, ensuring that generated solutions reflect both correctness and confidence.

## 4 EXPERIMENTS

In this section, we present the experimental setup used to assess CASPO and compare it with models trained by other state-of-the-art methods. In our main evaluation, we use consistent sampling settings across all reasoning models, with the default decoding parameters. More Training budgets (steps, batch size, total tokens, learning-rate schedule, and optimizer hyperparameters) detailed can be seen in Appendix A.

### 4.1 EXPERIMENTAL SETTINGS

**Models.** We conducted experiments with three open-source models: Llama-3.1-8B-Instruct (Grattafiori et al., 2024), Qwen2.5-Math-7B and Qwen2.5-7B-Instruct (Yang et al., 2024), which serve as our primary base models. To provide performance upper bounds, we further report results from large-scale models including GPT-4o (Jaech et al., 2024), o1-mini (o1 Team, 2024), and Claude-3.5-Sonnet (Anthropic, 2024), which are known for their strong reasoning capabilities. For answer calibration in training data, Qwen2.5-Math-7B-Instruct was used as the evaluator.

**Baselines.** We consider two categories of baselines: (i) *training-based methods*, which update model parameters from verifiable self-improvement signals, and (ii) *inference-time strategies*, which adjust decoding without modifying parameters.

**Training-based methods (Table 1).** We compare CASPO with four representative training-based methods that update model parameters from self-improvement signals, all relying on verifiable rewards but without additional SFT data. GRPO (Shao et al., 2024) is an off-policy group-relative policy optimization method: it leverages stored rollouts to compute verifiable rewards and optimizes relative advantages within a batch. Simple-RL-Zero (Zeng et al., 2025) adopts a lightweight PPO-style reinforcement learning setup trained solely with verifiable rewards; despite its simplicity, it still requires repeated on-policy sampling and updates, and the "Zero" denotes versions released at the time points. PURE-VR (Cheng et al., 2025) extends reinforcement learning to the trajectory level by propagating verifiable rewards across reasoning steps through a minimal credit assignment scheme. Finally, DPO-VP (Tu et al., 2025) applies iterative direct preference optimization on verifiable pairs of correct versus incorrect outputs, aligning the model policy through offline likelihood-based preference updates. These methods all treat reasoning at the trajectory level: they optimize which final answer is preferred but do not explicitly regulate the correctness or confidence of intermediate steps.

**Inference-time strategies (Table 2).** We also compare against reasoning-time methods that leave model parameters unchanged but influence the generation process. CoT (Kojima et al., 2022) unfolds intermediate reasoning steps; Self-Consistency (Wang et al., 2022) samples multiple reasoning chains and aggregates answers by majority vote; DiPT (Just et al., 2024) leverages diverse prompt

Table 1: Training-based comparison. Detailed performance of `CASPO` versus training-based baselines (GRPO (Guo et al., 2025), Simple-RL-Zero (Zeng et al., 2025), PURE-VR (Cheng et al., 2025), DPO-VP (Tu et al., 2025)) across multiple math benchmarks and base models. AIME24 is reported using two metrics: Pass@1 (single run) and Avg@32 (average score from 32 runs). AMC23 has 40 test items (2.5% per item).

| Models | Math 500 | Minerva Math | Olympiad Bench | AIME24 (Avg@ 1 / 32) | AMC23 | Avg |
|---|---|---|---|---|---|---|
| Claude-3.5-Sonnet | 78.3 | - | 32.5 | 16.0 | 57.5 | 46.1 |
| GPT-4o | 76.6 | - | 43.3 | 9.3 | 47.5 | 44.2 |
| o1-mini | 90.0 | - | 65.3 | 56.7 | 95.0 | 76.8 |
| Qwen2.5-Math-7B-Instruct | 76.4 | 33.5 | 38.4 | 20.0 | 62.5 | 46.2 |
| Qwen2.5-Math-7B | 64.8 | 15.4 | 25.6 | 16.7 | 37.5 | 32.0 |
| + GRPO | 76.2 | 32.7 | 38.1 | 16.7 | 55.0 | 43.7 |
| + Simple-RL-Zero | 78.0 | 33.1 | 36.6 | 20.0 | 57.5 | 45.0 |
| + PURE-VR | 79.8 | 36.8 | 41.9 | 20.0 | 57.5 | 47.5 |
| + DPO-VP | 74.8 | 35.3 | 36.9 | 23.3 | 60.0 | 46.1 |
| + CASPO | 76.6 | 37.8 | 43.8 | 23.3 | 62.5 | 48.8 |
| Qwen2.5-7B-Instruct | 76.2 | 37.6 | 43.0 | 13.3 | 52.5 | 44.4 |
| + GRPO | 79.0 | 41.0 | 46.5 | 13.3 | 55.0 | 46.6 |
| + Simple-RL-Zero | 80.2 | 41.5 | 45.8 | 16.7 | 57.5 | 47.8 |
| + PURE-VR | 81.5 | 43.0 | 47.5 | 16.7 | 57.5 | 48.9 |
| + DPO-VP | 79.8 | 42.5 | 46.2 | 20.0 | 60.0 | 49.1 |
| + CASPO | 82.0 | 44.0 | 48.3 | 20.0 | 62.5 | 50.6 |
| Llama-3.1-8B-Instruct | 49.6 | 13.2 | 23.5 | 6.7 | 27.5 | 24.1 |
| + GRPO | 52.0 | 15.0 | 25.0 | 6.7 | 30.0 | 25.5 |
| + Simple-RL-Zero | 53.2 | 15.5 | 25.6 | 10.0 | 30.0 | 26.9 |
| + PURE-VR | 54.0 | 16.0 | 26.8 | 10.0 | 32.5 | 27.6 |
| + DPO-VP | 54.8 | 16.5 | 27.2 | 13.3 | 32.5 | 28.8 |
| + CASPO | 55.2 | 15.6 | 27.6 | 13.3 | 35.0 | 29.1 |

templates to enhance coverage. While these methods improve robustness, they operate without explicit confidence calibration and may retain logically inconsistent reasoning paths.

Our method differs from both categories by unifying them into a *two-stage framework*. At the training stage, `CASPO` introduces step-wise confidence-aware preference optimization, leveraging both correct-but-uncertain and confidently-incorrect predictions as informative signals to align confidence with correctness at the step-wise. At the inference stage, it incorporates *CaT*, which dynamically expands or prunes reasoning trajectories according to cumulative step-wise confidence. Together, these two stages ensure that improvements accumulate across intermediate steps during training and are faithfully exploited at inference, achieving both stronger supervision and more reliable search than prior baselines.

**Evaluation and Benchmarks.** Our main evaluation focuses on mathematical reasoning benchmarks widely used in prior research, including MATH500 (subset of the MATH test set) (Lightman et al., 2023), Minerva-Math (Lewkowycz et al., 2022), and OlympiadBench (He et al., 2024), as well as competition-level benchmarks AMC2023 (MAA, a) and AIME2024 (MAA, b).

Beyond mathematics, we also examine the general reasoning transferability of the models on diverse out-of-domain benchmarks. These cover board game reasoning (BoardgameQA, BGQA) (Kazemi et al., 2023), code reasoning (CRUXEval, CRUX) (Gu et al., 2024), commonsense reasoning (StrategyQA, STGQA) (Geva et al., 2021), tabular reasoning (TableBench) (Wu et al., 2025), and STEM-specific reasoning (STEM subsets of MMLU-Pro) (Wang et al., 2024). This suite spans multiple disciplines, including physics, chemistry, computer science, engineering, biology, and economics, ensuring that `CASPO` is evaluated not only on mathematical tasks but also across a broad spectrum of reasoning challenges. More detailed can be seen in Appendix A.2.

Table 2: Inference-time comparison. Results of different inference strategies (CoT (Wei et al., 2022; Kojima et al., 2022), Self-Consistency(Wang et al., 2022), DiPT (Just et al., 2024), CaT) applied on top of CASPO-trained models. All methods are evaluated on the same math benchmarks with identical training; only inference-time strategies differ.

| Models | Math 500 | Minerva Math | Olympiad Bench | AIME24 (Avg@1/32) | AMC23 | Avg |
|---|---|---|---|---|---|---|
| Qwen2.5-Math-7B-CASPO | 76.6 | 37.8 | 43.8 | 23.3 | 62.5 | 48.8 |
| + CoT | 78.2 | 38.6 | 44.7 | 23.3 | 63.8 | 49.7 |
| + Self-Consistency | 79.6 | 39.3 | 45.6 | 26.7 | 65.0 | 51.2 |
| + DiPT | 80.0 | 39.5 | 45.8 | 23.3 | 65.0 | 50.7 |
| + CaT (Ours) | 81.9 | 40.5 | 46.9 | 26.7 | 67.5 | 52.7 |
| Qwen2.5-7B-Instruct-CASPO | 82.0 | 44.0 | 48.3 | 20.0 | 62.5 | 50.6 |
| + CoT | 83.6 | 44.9 | 49.3 | 20.0 | 63.8 | 52.3 |
| + Self-Consistency | 85.3 | 45.8 | 50.2 | 23.3 | 65.0 | 53.9 |
| + DiPT | 85.7 | 46.0 | 50.5 | 20.0 | 65.0 | 53.4 |
| + CaT (Ours) | 87.7 | 47.1 | 51.7 | 26.7 | 67.5 | 56.1 |
| Llama-3.1-8B-Instruct-CASPO | 55.2 | 15.6 | 27.6 | 13.3 | 35.0 | 29.1 |
| + CoT | 56.3 | 15.9 | 28.1 | 13.3 | 36.3 | 30.0 |
| + Self-Consistency | 57.4 | 16.2 | 28.7 | 16.7 | 37.5 | 31.3 |
| + DiPT | 57.7 | 16.3 | 28.8 | 13.3 | 37.5 | 30.7 |
| + CaT (Ours) | 59.1 | 16.7 | 29.5 | 20.0 | 40.0 | 33.1 |

## 4.2 MAIN RESULTS

**Training-based comparison.** Table 1 compares CASPO with training baselines under matched training and inference budgets. Across all three bases, CASPO delivers consistent gains; e.g., with **Qwen2.5-7B-Instruct** it reaches an Avg of **50.6**, surpassing GRPO, Simple-RL-Zero, PURE-VR, and DPO-VP. These improvements arise from *step-wise confidence-aware preference learning*, which better aligns probabilities with intermediate-step correctness. Figure 5 shows accuracy trends: despite occasional noise from ambiguous items, the overall trajectory is monotonic, indicating stable self-improvement as calibration accumulates.

**Inference-time comparison.** Table 2 evaluates inference strategies on *CASPO-trained* models under the same sampling budget ($K{=}10$). The number of generated path $K$ is set to 10, the choice supported by previous research (Zhang et al., 2023; Duan et al., 2023; Qiu & Miikkulainen, 2024; Fadeeva et al., 2024; Razghandi et al., 2025).Both Self-Consistency and CaT yield larger deltas on CASPO models than on instruct-tuned counterparts, showing thattraining-time calibration transfers to inference-time search. Specifically, CaT achieves the best averages across bases while keeping the sampling budget fixed.

**Out-of-Domain Transferability** Although CASPO is trained on mathematical data, it transfers reliably to non-math reasoning under the same prompts and sampling budget. As shown in Table 3, CASPO consistently improves performance across diverse benchmarks, including commonsense reasoning (STGQA), code reasoning (CRUX), tabular reasoning (TableBench), and STEM knowledge (MMLU-Pro STEM). For MMLU-Pro STEM, Across 6 STEM subsets (physics, chemistry, computer science, engineering, biology, economics; 5,371 problems in total), CASPO improves Qwen2.5-Math-7B from 52.8 to 57.2, Qwen2.5-7B-Instruct from 58.2 to 61.6, and Llama-3.1-8B-Instruct from 47.6 to 50.5. These gains demonstrate that step-wise aggregation generalizes beyond mathematics, providing a lightweight yet robust inference-time strategy for diverse reasoning challenges.

## 4.3 ANALYSIS

**Training Dynamics.** Following prior work on DPO training dynamics (Ren & Sutherland, 2024), we examine reward evolution during optimization. As shown in Figure 2, all models exhibit the expected dual-pressure mechanism: chosen rewards initially drop before recovering near zero, while rejected rewards decrease monotonically, confirming the theoretical framework of simultaneous

Table 3: Consistent out-of-domain reasoning enhancement with `CASPO` : Achieving generalization at equivalent model scales beyond mathematical training.

| Models | BGQA | CRUX | STGQA | TableBench | MMLU-STEM | Avg |
|---|---|---|---|---|---|---|
| Qwen2.5-Math-7B | 48.0 | 50.0 | 88.0 | 38.0 | 40.0 | 52.8 |
| + GRPO | 50.5 | 53.0 | 89.5 | 39.0 | 42.0 | 54.8 |
| + Simple-RL-Zero | 51.5 | 53.5 | 90.0 | 40.0 | 42.5 | 55.5 |
| + PURE-VR | 52.0 | 54.0 | 90.5 | 40.5 | 43.0 | 56.0 |
| + DPO-VP | 52.5 | 54.5 | 91.0 | 41.0 | 43.5 | 56.5 |
| + `CASPO` | 53.5 | 55.5 | 91.5 | 41.5 | 44.0 | 57.2 |
| + `CASPO` + CaT | 56.2 | 58.3 | 96.1 | 43.6 | 46.2 | 60.1 |
| Qwen2.5-7B-Instruct | 53.0 | 58.1 | 91.3 | 43.2 | 45.2 | 58.2 |
| + GRPO | 54.5 | 59.9 | 92.1 | 44.0 | 46.2 | 59.3 |
| + Simple-RL-Zero | 55.5 | 60.9 | 92.5 | 44.4 | 46.7 | 60.0 |
| + PURE-VR | 56.0 | 61.4 | 92.8 | 44.7 | 47.0 | 60.4 |
| + DPO-VP | 56.8 | 62.1 | 93.3 | 45.2 | 47.4 | 61.0 |
| + `CASPO` | 57.5 | 62.9 | 93.8 | 45.7 | 48.0 | 61.6 |
| + `CASPO` + CaT | 60.4 | 66.0 | 98.5 | 48.0 | 50.4 | 64.7 |
| Llama-3.1-8B-Instruct | 40.0 | 45.0 | 82.0 | 35.0 | 36.0 | 47.6 |
| + GRPO | 41.5 | 46.5 | 83.0 | 35.8 | 37.0 | 48.8 |
| + Simple-RL-Zero | 42.0 | 47.0 | 83.5 | 36.2 | 37.2 | 49.2 |
| + PURE-VR | 42.5 | 47.5 | 83.8 | 36.5 | 37.5 | 49.6 |
| + DPO-VP | 43.0 | 48.0 | 84.2 | 37.0 | 38.0 | 50.0 |
| + `CASPO` | 43.5 | 48.5 | 84.5 | 37.5 | 38.5 | 50.5 |
| + `CASPO` + CaT | 45.7 | 51.0 | 88.8 | 39.4 | 40.4 | 53.1 |

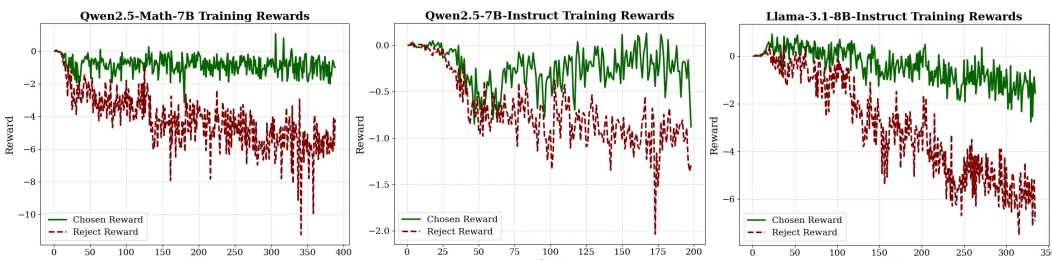

Figure 2: Reward evolution during DPO training across Qwen2.5-Math-7B, Qwen2.5-7B-Instruct, and Llama-3.1-8B-Instruct models

upward and downward pressures. For completeness, we also examine the evolution of training accuracy and loss, with the corresponding curves provided in Appendix C.

Our results reveal clear model-specific patterns: Qwen2.5-Math-7B converges the fastest and with the greatest stability, achieving the largest reward separation of about 6.0. This reflects its strong alignment with mathematical reasoning preferences, strengthened by domain-specific pre-training. Qwen2.5-7B-Instruct converges efficiently within 200 steps, reaching a moderate separation of about 1.5, which indicates a balance between training efficiency and preference learning. Llama-3.1-8B-Instruct shows higher volatility during optimization but eventually achieves a separation comparable to the Math model, although this requires more careful tuning of hyperparameters.

The evolution of training accuracy and loss closely mirrors the reward dynamics. As chosen rewards recover and separate from rejected ones, accuracy steadily increases and stabilizes at a high level, indicating that the models are learning to generate consistently preferred responses. At the same time, the training loss decreases in a smooth downward trajectory, with sharper drops aligning with phases of rapid accuracy improvement. These observations reinforce the interpretation that reward separation not only drives preference alignment at the signal level but also translates into tangible accuracy gains and more confident optimization. Together, the three perspectives—reward, accuracy, and loss—offer a coherent view of how DPO training progresses and validate the robustness of our implementation across different base models.

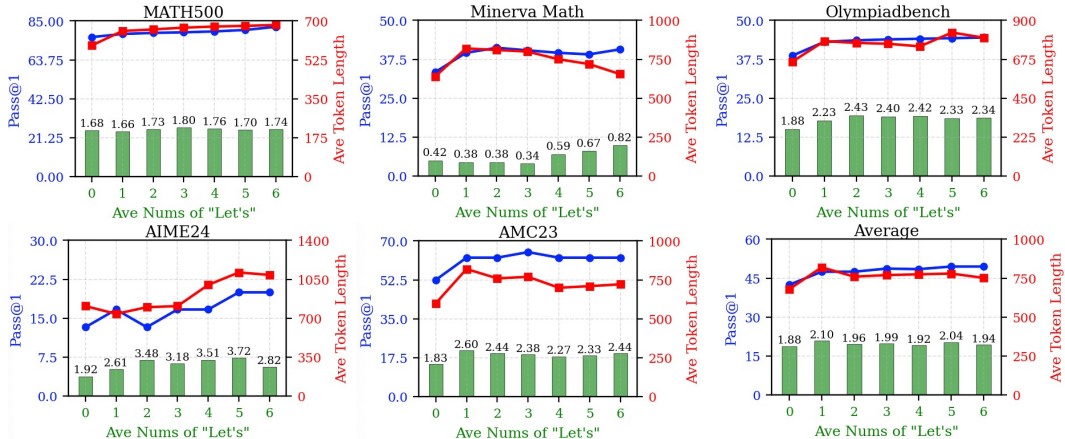

Figure 3: Pass@1 accuracy improves consistently across DPO rounds without a substantial increase in token length. Meanwhile, the use of self-talk triggers (Let's) declines or stabilizes, suggesting that DPO guides models toward more concise and targeted reasoning.

**Token Length and Reasoning Pattern Evolution.** Across multiple rounds of DPO training, Qwen2.5-7B-Instruct shows steadily increasing Pass@1 accuracy on math benchmarks, while the overall token length of generated solutions remains relatively stable, indicating that performance gains are not driven by longer reasoning chains. To capture shifts in reasoning task, we follow prior studies (Zhou et al., 2025) and use the frequency of the self-talk trigger (let's) as a proxy (Tu et al., 2025) for explicit self-checking. As shown in Figure 3, we observe a gradual decline in its occurrence, suggesting that the model reduces redundant self-reflection and instead relies more directly on preference signals optimized by DPO to produce correct answers. This pattern implies that training does not create a new reflective ability but rather strengthens the model's existing tendency to self-check, enabling it to achieve higher-quality reasoning with fewer triggers and without additional inference cost.

## 4.4 ABLATIONS

**Iterative Training.** To understand the effect of iterative training, we let CASPO run for three rounds, where in each round the current best model generates fresh step-wise data for the next update. This setup differs from standard multi-epoch fine-tuning, since the data distribution itself evolves with the model. As shown in Figure 4, all benchmarks improve steadily across the three iterations. The first round contributes the largest boost—for example, Math500 jumps from 64.8 to 76.6 and AMC23 rises from 37.5 to 60.0 in a single step. Later rounds still bring non-trivial but smaller gains, such as OlympiaBench frrom 37.8 to 38.7, and AMC23 from 60.0 to 62.5. This pattern suggests a natural dynamic: the first iteration corrects the most obvious miscalibrations, while subsequent iterations act more like refinements, gradually aligning confidence

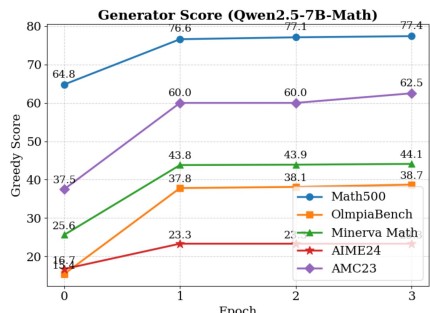

Figure 4: Effect of Iterative Training with CASPO on Qwen2.5-7B-Math across Multiple Benchmarks.

with reasoning quality. Overall, the results support our claim that step-wise iterative self-generation creates a positive feedback loop—better models produce better training data, and better data in turn makes the models stronger.

**Balance between Diversity and Reliability.** The ablation studies complement the comparisons in Table 2 by highlighting how CASPO reshapes the balance between diversity and reliability. Models trained with CASPO already benefit from more accurate confidence signals, which makes the candidates generated under pass@k sampling more dependable. Consequently, when diversity is introduced through sampling, the resulting pool of answers contains less noise and better reflects the model's calibrated preferences. On top of this, aggregation strategies such as majority voting or

Self-Consistency further amplify the advantage: instead of averaging over unreliable candidates, they combine multiple high-quality and well-calibrated reasoning paths. This explains why methods like CaT deliver consistently stronger gains when applied to `CASPO` -trained models, not only improving average performance but also enhancing stability across benchmarks.

## 5 CONCLUSION

This work investigates the limitations of large language models in generating faithful intermediate reasoning, where correct final answers may sometimes result from logically inconsistent steps. We introduce `CASPO`, a framework that unifies confidence-aware preference optimization in training with confidence-guided self-revision at inference, aligning step-wise confidence with correctness and using it to refine reasoning trajectories. Experimental results on mathematical and out-of-domain benchmarks demonstrate that `CASPO` consistently enhances both accuracy and reasoning faithfulness compared to strong baselines, indicating its potential to support further advances in multi-step reasoning with LLMs.

**Limitations.** While `CASPO` shows consistent gains across benchmarks and model families, it also has limitations. It currently relies on an external evaluator for step-wise verification, which may introduce bias, and its entropy-based confidence estimation may not fully capture uncertainty. Future work could explore self-contained verification and more advanced calibration methods to further improve robustness and generality.

## ETHICAL STATEMENT

Our method enhances reasoning performance through step-wise optimization, enabling language models to generate more reliable and interpretable solutions. While this contributes positively to building trustworthy AI systems, stronger reasoning ability may also amplify risks if misapplied in sensitive domains. We therefore encourage researchers and practitioners to adopt `CASPO` responsibly, with careful consideration of downstream applications and potential societal impacts.

## LARGE LANGUAGE MODEL USAGE

During the preparation of this manuscript, LLMs were employed exclusively for editorial purposes, including language polishing, grammar correction, and style refinement. Their use was restricted to improving readability and presentation. LLMs did not contribute to the conception of ideas, experimental design, analysis, or interpretation of results. All scientific content is solely the work of the authors.

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

In this appendix, we provide comprehensive information on the Experimental Setup A, Critic Evaluation Process B, and Additional Results C.

# A EXPERIMENTAL SETUP

## A.1 DETAILS OF TRAINING AND EVALUATION

The base models include Qwen2.5-Math-7B, Qwen2.5-7B-Instruct, and Llama-3.1-8B-Instruct. All model-centric training was conducted with full-parameter fine-tuning using the Open-RLHF framework Hu et al. (2024). Random seeds are fixed at 42 for reproducibility. All experiments are trained on 4 NVIDIA A800 GPUs (80GB) with mixed-precision (FP16) enabled.

**Optimization hyperparameters.** The SFT stage uses a learning rate of $5 \times 10^{-6}$, while the Direct Preference Optimization (DPO) stage adopts $5 \times 10^{-7}$ to stabilize preference-based updates. Both stages share a maximum sequence length of 2048 tokens and a batch size of 64. The DPO loss coefficient $\beta$ is fixed at 0.1. For each DPO round, candidate responses were sampled with temperature $t = 0.7$, and preference pairs were filtered according to the verifiable-pair criterion described in Section 3.2.

**Training schedule.** Each training run lasts six epochs. For the first three epochs, the sampling temperature is fixed at $t = 0.7$ to keep the data distribution close to the initial policy. For epochs four and five, it is increased to $t = 1.0$, and further raised to $t = 1.2$ in the final epoch. This annealed schedule reflects the observation that performance plateaus after three epochs, while higher temperatures promote exploration of novel reasoning paths without destabilizing optimization.

**Evaluation protocol.** We follow the Qwen-Math evaluation suite. For every benchmark and model, generations are produced with greedy decoding ($t$=0.0), one output per input (no sampling, no self-consistency), and a 2048-token generation limit. All models use the same zero-shot CoT prompt template (shown below) to avoid prompt-engineering confounds. We report pass@1. For datasets that provide official scoring scripts, we use those scripts; otherwise, answers are extracted from the boxed span (see below) and matched after standard normalization.

```
Prompt Template:

<|im_start|>user
Solve the following math problem efficiently and clearly.
Please reason step by step, and put your final answer within {\boxed{}}.
Problem:  «<your instruction»>
<|im_end|>

<|im_start|>assistant
```

## A.2 DETAILS OF BENCHMARKS

**MATH500** (Lightman et al., 2023) is a subset of the MATH benchmark (Hendrycks et al., 2021), containing 500 uniformly sampled problems. Its distribution of subjects and difficulty levels is representative of the full MATH test set, making it a widely adopted evaluation for mathematical reasoning.

**Minerva-Math** (Lewkowycz et al., 2022) is a high-difficulty math problem dataset consisting of 272 challenging problems. Some problems are also relevant to scientific topics in other subjects, such as physics.

**OlympiadBench** (He et al., 2024) is a bilingual, multimodal benchmark featuring 8,476 Olympiad-level math and physics problems, including problems adapted from the Chinese college entrance exam. We use the text-only, open-ended math competition subset, which contains 674 problems.

**AMC2023** and **AIME2024** are competition-style benchmarks. AMC2023 includes 40 text-only problems from the 2023 American Mathematics Competition, while AIME2024 contains 30 text-only problems from the 2024 American Invitational Mathematics Examination.

**BoardgameQA (BGQA)** (Kazemi et al., 2023) is a logical reasoning dataset with 15K unique problems designed to test LLMs' ability to handle defeasible reasoning, where contradictions are resolved via credibility or recency cues.

**CRUXEval** (Gu et al., 2024) evaluates code reasoning and execution. It contains 800 Python functions (3–13 lines) with input–output pairs, where models must generate the correct outputs given a function snippet and input example.

**StrategyQA** (Geva et al., 2021) contains 2,780 multi-hop reasoning questions where the reasoning steps are implicit and must be inferred using a strategy. Each example is paired with a decomposition into sub-steps and supporting evidence from Wikipedia.

**TableBench** (Wu et al., 2025) evaluates tabular reasoning in real-world data analysis tasks across 18 domains. We use the subsets on fact-checking and numerical reasoning, resulting in 491 unique problems. Tasks cover fact validation, numerical calculation, and reasoning over structured tables.

**MMLUPro-STEM** (Wang et al., 2024) is a subset of MMLU-Pro, an enhanced version of MMLU (Hendrycks et al., 2020) with more reasoning-focused questions and expanded answer choices (four to ten). We select six STEM domains—physics, chemistry, computer science, engineering, biology, and economics—excluding math as it overlaps with in-domain tasks. The final test set contains 5,371 unique problems.

### A.3 STEP-WISE AGGREGATION FUNCTION CHOICE

In `CASPO`, we define a step-wise aggregation function $f$ that quantifies the confidence of a step-level answer by leveraging the probabilities of its constituent tokens. Intuitively, this function serves as a bridge from token-level uncertainty to step-level confidence. We consider two common formulations for $f$:

- **Mean Entropy.** Compute the average entropy of all tokens in the answer. A lower entropy indicates higher confidence, as the model is more certain about its token predictions:

$$f_{\text{entropy}}(s) = -\frac{1}{n} \sum_{i=1}^{n} \sum_{v \in \mathcal{V}} p(t_i = v) \log p(t_i = v). \tag{5}$$

- **Multiplicative Probability.** Compute the joint probability of the entire answer by multiplying the probabilities of its tokens. A higher value indicates greater confidence in the step:

$$f_{\text{mult}}(s) = \prod_{i=1}^{n} p(t_i). \tag{6}$$

We also conduct ablation experiments to compare these choices, and find that both metrics capture different aspects of confidence: mean entropy emphasizes uncertainty calibration, while multiplicative probability reflects overall likelihood of a step.

### A.4 CASPO ALGORITHM

The algorithm explicitly describes how step-wise answers are generated, confidence is estimated, and preference pairs are constructed for Direct Preference Optimization. In particular, it clarifies how both "correct but low-confidence" predictions and "confidently incorrect" predictions are incorporated into the preference dataset, which is then used in the iterative DPO optimization loop. This pseudocode is intended to complement the main text by offering an implementation-oriented view of our framework

**Algorithm 1** CASPO

**Input:** Dataset with math questions: $\mathcal{D}_{\text{math}} = \{x\}$; target model $\pi_\theta$; target model tokenizer $\mathcal{T}_\theta$; alignment hyper-parameter $\beta$; Iterations: $K$.

1: $\mathcal{D} \leftarrow \{\}$ // Construct preference dataset $\mathcal{D}$ for training.
2: **for** each question $x$ in $\mathcal{D}_{\text{math}}$ **do**
3:   **for** $j = 1, \ldots, m$ **do**
4:     $\mathbb{P}[\mathcal{T}(\mathcal{V})] \leftarrow \pi_\theta(\cdot|q_j)$   // Generate answer for the sub question $q_j$.
5:     $s_j \leftarrow$ top-1 token with highest likelihood (greedy decoding).
6:     $c_j \leftarrow \text{confidence}(s_j)$   // Get the confidence of answer $s_j$ with Eq.1
7:     $g_j \leftarrow \pi_{\text{critic}}(\cdot|q_j)$ // Get the ground true answer from the critic model.
8:     **if** $s_j == g_j$ **then**
9:       **if** $c_j > \tau$ **then**
          Continue
10:      **else**
11:        $y_w \leftarrow s_j$ Correct but low confidence.
12:        $y_l \leftarrow$ the token with the second highest likelihood.
13:      **end if**
14:    **else**
15:      $y_w \leftarrow g_j$
16:      $y_l \leftarrow s_j$
17:    **end if**
18:    $\mathcal{D} = \mathcal{D} \cup \{q_j, y_j^w, y_j^l\}$
19:   **end for**
20: **end for**
21: **for** $k = 1, \ldots, K$ **do**
22:   $\pi_{\theta_k} \leftarrow \arg\min_\theta \mathcal{L}(\pi_{\theta_k}, \pi_{\text{ref}})$ //Update target model.
23:   $\pi_{\text{ref}} \leftarrow \pi_{\theta_k}$     //Update reference model.
24: **end for**

# B  CRITIC EVALUATION PROMPT

**Critic verifies correctness**

```
## General Guidelines
You are a student. Your task is to carefully review the correct
partial solution to a math problem, and adhere to the
following guidelines:

1. Verify the correctness of the solution and explain the reason:
"Verify: [Explanation of why you are correct with one sentence]"

2. You are provided with the question, the ground truth solution,
and your step-by-step partial solution.

3. Your response should be exactly in the following format: Verify:
[brief explanation of why you are correct with one sentence]

## Test Example
### Question
<<<question>>>

### Ground truth solution
<<<gt_solution>>>

### Your partial solution
<<<student_solution>>>

### Your review
```

| | **Math** | | | **Open-domain** |
| LLM | Math500 | Minerva Math | OlympiadBench | MMLU-STEM |
|---|---|---|---|---|
| *Multiplication* | | | | |
| Qwen2.5-Math-7B | 63.2 | 14.7 | 24.9 | 41.9 |
| Qwen2.5-7B-Instruct | 80.5 | 32.7 | 38.1 | 45.2 |
| Llama3.1-8B-Instruct | 48.7 | 12.8 | 22.6 | 36.1 |
| *Entropy* | | | | |
| Qwen2.5-Math-7B | 64.8 | 15.4 | 25.6 | 42.5 |
| Qwen2.5-7B-Instruct | 83.2 | 33.5 | 38.4 | 45.6 |
| Llama3.1-8B-Instruct | 49.6 | 13.6 | 23.5 | 36.0 |

Table 4: Accuracy comparison of LRMs on mathematical reasoning and open-domain datasets. The math datasets include Math500, Minerva Math, and OlympiadBench; the open-domain dataset is MMLU-STEM. Results are reported for two variations of the step-wise aggregate function $f$: Multiplication and Entropy.

## C  ADDITIONAL RESULTS

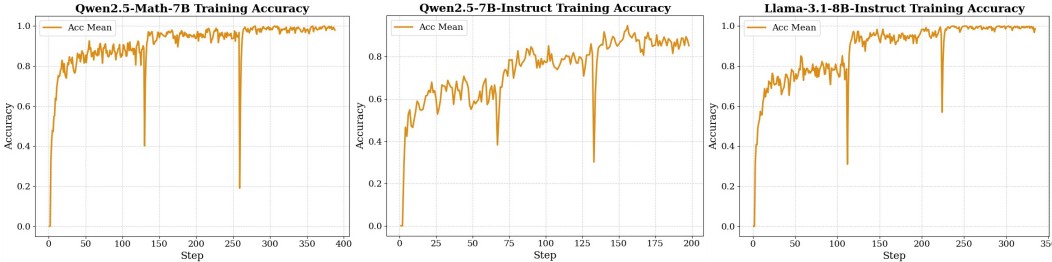

Figure 5: Training accuracy trajectories of Qwen2.5-Math-7B, Qwen2.5-7B-Instruct, and Llama-3.1-8B-Instruct across different optimization steps

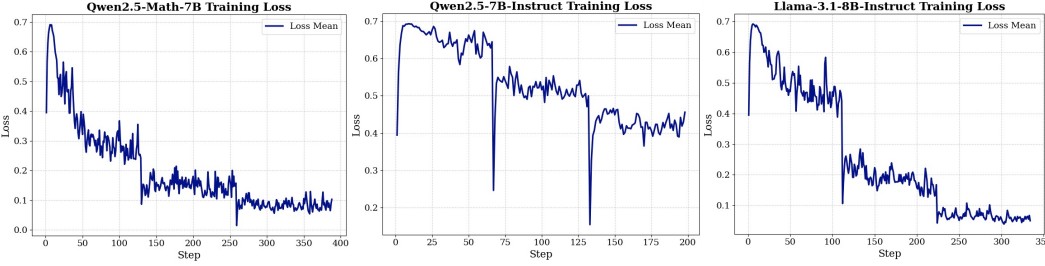

Figure 6: Loss reduction patterns for Qwen2.5-Math-7B, Qwen2.5-7B-Instruct, and Llama-3.1-8B-Instruct during DPO training optimization process

**Accuracy and Loss Dynamics.**  Figures 5 and 6 illustrate how accuracy and loss evolve alongside reward dynamics. Across all models, training accuracy improves steadily as reward separation emerges, ultimately stabilizing at high levels. Loss curves show a consistent downward trend, with sharper decreases during phases of rapid accuracy gains. Together, these two signals provide complementary evidence that preference learning is not only separating rewards but also improving the reliability of correct predictions.

Qwen2.5-Math-7B again shows the smoothest trajectory, with accuracy quickly approaching near-perfect levels and loss decreasing steadily. Qwen2.5-7B-Instruct reaches stable accuracy around 0.9 within 200 steps, accompanied by moderate but consistent loss reduction. Llama-3.1-8B-Instruct improves more slowly and exhibits more fluctuations in loss, but ultimately converges to strong accuracy with sufficiently long training.

**step-wise aggregate funtion** $f$. To construct the confidence of a reasoning path, we need to aggregate the step-wise confidences into a single score. We consider two natural choices for the aggregate function $f$: (i) Multiplication, here the cumulative confidence of a path is obtained by multiplying the confidence values across all steps. This strict formulation penalizes any step with low confidence, effectively ensuring that only trajectories with uniformly reliable reasoning are preserved. (ii) Entropy-based aggregation, where the token-level entropy across steps is averaged to measure the overall uncertainty of a trajectory. This softer formulation emphasizes calibration by capturing the global uncertainty pattern, rather than enforcing per-step reliability.

Table 4 compares the two approaches. Multiplication favors precision by aggressively pruning uncertain branches, while entropy aggregation provides more tolerant coverage and may preserve diverse but moderately confident paths. The empirical results highlight the trade-off: multiplication often achieves higher accuracy on deterministic math tasks, whereas entropy can be advantageous when some steps carry inherent ambiguity, as in open-domain settings.

