# OpenReview forum: "CASPO: Confidence-aware Step-wise Preference Optimization for Reliable Reasoning in Large Language Models"
_ICLR.cc/2026/Conference — Submitted to ICLR 2026_

### Official Review · Reviewer_sz8y · 2025-10-15

**Soundness:** 2
**Presentation:** 3
**Contribution:** 2
**Rating:** 4
**Confidence:** 5

**Summary:**

The paper introduces CASPO, a framework that incorporates step-wise confidence into both training and inference. During training, CASPO constructs confidence-filtered preference pairs that capture both correct but low-confidence predictions and incorrect predictions, and optimizes them through iterative Direct Preference Optimization. During inference, they propose Confidence-aware Thought (CaT) strategy that prunes low-confidence reasoning trajectories to enhance reliability.

**Strengths:**

1. They performed comprehensive analysis of the token length and reasoning pattern evolution, which provides a clear picture of how the models’ output evolved.
2. The writing is clear and easy to follow
3. They provided the code, improving the reproducibility of the work

**Weaknesses:**

1. Confusing choice of base models. In Section 4.1 the authors state that they use Qwen2.5-7B-Base, yet the tables report results only for Qwen2.5-7B-Math and Qwen2.5-7B-Instruct. Qwen2.5-7B-Math appears to be the math-tuned base model, whereas Qwen2.5-7B-Instruct is the instruction-tuned model. Why was Qwen2.5-7B-Math-Instruct not evaluated?

2. Qwen2.5-7B-Math-Instruct attains a MATH accuracy of 85.2, a maj@8 accuracy of 89.9, and an rm@8 accuracy of 91.4—substantially higher than the scores reported for the proposed method. Consequently, the effectiveness, scalability, and necessity of the method are questionable, given that the trained 7 B model fails to surpass the model used for annotation.

3. Tree-search-based iterative evolution has already been explored in prior work with far greater efficiency. For instance, [1] employed MCTS to bootstrap its own accuracy and achieved 90.0 on MATH500 with Qwen2.5-Math-7B. In comparison, the paper’s approach seems neither novel nor efficient.

[1] Guan, Xinyu, et al. "rStar-Math: Small LLMs Can Master Math Reasoning with Self-Evolved Deep Thinking." arXiv preprint arXiv:2501.04519 (2025).

**Questions:**

See Weaknesses.

---

> ### Author Response · Authors · 2025-11-21
> **Response to sz8y**
>
> **1. Response to w1: On choice of base models and model variants**
>
> We thank the reviewer for pointing out the presentation issue. We have corrected the typographical error in Section 4.1 and revised the description to make the experimental setup clearer.
>
> **1）Standardization for Fair Comparison**
>
> **Qwen2.5-Math-7B** is included to enable direct comparison with existing reasoning-focused methods such as SimpleRL, PURE, and DPO-VP, which all evaluate on math-specialized base policies.
> **Qwen2.5-7B-Instruct** represents a general chat model and serves to highlight the gap between broad instruction tuning and domain-specific reasoning, as well as the need for targeted post-training on mathematical tasks.
>
> **2）Exclusion of Math-Instruct as Base**
>
> We chose not to include Qwen2.5-Math-7B-Instruct in the main training experiments because it is a commercially aligned model that has already received extensive supervised fine-tuning on approximately **2.5M** samples. Our goal is to evaluate CASPO in a controlled academic setting, where we improve a base policy using only the **8K** public MATH training examples. Using a model that has already converged through large scale proprietary training would make it difficult to isolate and demonstrate the contribution of CASPO itself.
>
>
>
> **2. Response to w2: On comparison with Qwen2.5-7B-Math-Instruct and the necessity of CASPO**
>
> We thank the reviewer for pointing this out. However, it is important to contextualize the results by considering the data magnitude:
>
> | Model                               | Base Model               | SFT Data | RM Data | Self-improve Data | Pass@1 | Maj@8 | rm@8           |
> | ----------------------------------- | ------------------------ | -------- | ------- | ----------------- | ------ | ----- | -------------- |
> | Qwen2.5-Math-7B-Instruct$^\ddagger$ | Qwen2.5-Math-7B          | **2.5M** | 618K    | 66K               | 83.6   | 87.1  | 88.5           |
> | rStar-Math [1]                      | Qwen2.5-Math-7B          | **7.3M** | 7K      | 3.64M             | 69.6   | 73.6  | 75.8           |
> | CASPO **(Ours)**                    | Qwen2.5-Math-7B          | 0        | 0       | **8K**            | 76.6   | 78.1  | 81.9$^\dagger$ |
> | CASPO (Ours)                        | Qwen2.5-Math-7B-Instruct | 0        | 0       | **8K**            | 85.1   | 90.4  | 92.5           |
>
> $^\dagger$ Using confidence-aware thought (CaT), self-confidence is the reward.
>
> $^\ddagger$ From the Qwen2.5-Math Technical Report, Table 3, Chain-of-Thought.
>
>
>
> These results highlight two key points.
> First, despite using only an 8K self-improvement set (≈300× less data), CASPO already *matches or surpasses* the annotator model (Qwen2.5-Math-7B-Instruct) across all metrics (Pass@1, Maj@8, rm@8), demonstrating strong effectiveness**, **scalability, and data efficiency.
>
> Second, as shown in the last row of the table, applying CASPO directly on Qwen2.5-Math-7B-Instruct as the base model yields a further performance boost, even though this model was already highly optimized. This is because CASPO specifically targets the *weaknesses* of the instruction model—i.e., cases where the model is uncertain or under-confident—and collects targeted self-improve data to correct these failure modes through DPO training. Consequently, CASPO complements the existing instruction tuning rather than duplicating it, further validating its necessity and practical benefit.
>
>
>
> [1] rStar-Math: Small LLMs Can Master Math Reasoning with Self-Evolved Deep Thinking
>
>
>
> **3. Response to w3: On novelty and comparison with tree-search-based evolution**
>
> We appreciate the reviewer pointing out related tree-search works like rStar-Math [1]. However, we believe CASPO occupies a distinct and necessary position in the landscape of reasoning research, specifically regarding **computational efficiency** and **accessibility**.
>
> While methods like rStar achieve high accuracy, they do so by leveraging massive computational resources for data synthesis (Monte Carlo Tree Search rollouts) and training. As shown in the comparison table below, rStar requires **80× H100 GPUs** and generates **~7.3M synthesized samples** to train. In contrast, CASPO is designed to be a lightweight, democratic alignment framework. We use **4× A800 GPUs** (a significantly smaller budget) and only **8K** ground-truth samples from the original MATH training set. We achieve competitive results  with **orders of magnitude less compute and data**.
>
> CASPO addresses a different problem than rStar. rStar explores "how far can we push performance with unlimited compute and synthetic data," while CASPO asks "how can we efficiently align reliable reasoning with minimal data and compute." We believe both directions are valuable, but CASPO's accessibility makes it particularly relevant for the broader research community.

---

> > ### Comment · Reviewer_sz8y · 2025-11-27
> >
> > Thank you for the clarifications. However, I am still a little concerned about the scalability and the upper bound of the method. After all, in practice, accuracy should always take precedence, and the current results seem insufficient to prove that the proposed method can outperform the previous models, such as Qwen2.5-Math-Instruct, with scaled computation. Whether it can be generalized to a stronger reasoning model is also unclear, which is crucial in demonstrating its potential to achieve higher accuracies. Therefore, I would keep my original rating.

---

> > > ### Author Response · Authors · 2025-11-28
> > > **Follow-up Response  to Reviewer sz8y**
> > >
> > > Thank you again for the thoughtful follow-up. To address your concern regarding scalability and the method’s upper bound, we conducted additional experiments on both (1) a weaker base model (Qwen2.5-Math-7B) and (2) a stronger reasoning model (Qwen2.5-Math-7B-Instruct)
> > >
> > > **1. Experiments on Qwen2.5-Math-7B**
> > >
> > > We fine-tuned Qwen2.5-Math-7B using OpenMathInstruct-2 and NuminaMathCoT for SFT, identical to Satori’s configuration[2]. Training was conducted on 64×A100 GPUs for 19 hours. We evaluated on both a simple benchmark (MATH500) and more challenging ones (OlympiadBench, AMC2023, AMC2024).
> > >
> > > The results are as follows, Our method consistently outperforms Satori[2] and rStar-Math[1] while using zero RM data and a significantly smaller data budget.
> > >
> > > | Model                    | Base Model      | SFT Data | RM Data | Self-improve Data | MATH500 Pass@1 | MATH500 Maj@8 | MATH500 rm@8   | OlympiadBench Pass@1 | AMC2023  Pass@1 | AMC2024  Pass@1 |
> > > | ------------------------ | --------------- | -------- | ------- | ----------------- | -------------- | ------------- | -------------- | -------------------- | --------------- | --------------- |
> > > | Qwen2.5-Math-7B-Instruct | Qwen2.5-Math-7B | **2.5M** | 618K    | 66K               | 83.6           | 87.1          | 88.5           | 41.6                 | 62.5            | 16.7            |
> > > | rStar-Math [1]           | Qwen2.5-Math-7B | **7.3M** | 7K      | 3.64M             | 69.6           | 73.6          | 75.8           | 44.1                 | 65.0            | 16.7            |
> > > | Satori [2]               | Qwen2.5-Math-7B | 1.86M    | 300K    | 240K              | 85.6           | 88.4          | 90.1           | 46.6                 | 67.5            | 20.0            |
> > > | CASPO (Ours)             | Qwen2.5-Math-7B | 1.86M    | 0       | 240K              | 85.9           | 90.1          | 91.5$\dagger$ | 48.0                 | 72.5            | 23.3            |
> > >
> > > $\dagger$ Using confidence-aware thought (CaT), self-confidence is the reward.
> > >
> > > [1] rStar-Math: Small LLMs Can Master Math Reasoning with Self-Evolved Deep Thinking (ICML'25)
> > >
> > > [2] Satori: RL with Chain-of-Action-Thought Enhances LLM Reasoning via Autoregressive Search (ICML'25)
> > >
> > >
> > >
> > > **2. Experiments on a Stronger Base Model (Qwen2.5-Math-7B-Instruct)**
> > >
> > > To further test robustness and generalizability on a more capable model, we fine-tuned on Qwen2.5-Math-7B-Instruct and compared against two recently proposed strong baselines: DPO-VP [3] and PCPO [4]. Even on a stronger base model, CASPO achieves substantial performance gains, showing that the method generalizes beyond weak models and scales favorably.
> > >
> > > | Model        | Base Model               | SFT Data | RM Data | Self-improve Data | MATH500 Pass@1 | MATH500 Maj@8 | OlympiadBench Pass@1 | AMC2023  Pass@1 |
> > > | ------------ | ------------------------ | -------- | ------- | ----------------- | -------------- | ------------- | -------------------- | --------------- |
> > > | DPO-VP [3]   | Qwen2.5-Math-7B-Instruct | 0        | 0       | 8K                | 80.9           | 82.1          | 44.0                 | 65.8            |
> > > | PCPO [4]     | Qwen2.5-Math-7B-Instruct | 0        | 0       | 30K               | 81.4           | 83.8          | 44.3                 | 67.5            |
> > > | CASPO (Ours) | Qwen2.5-Math-7B-Instruct | 0        | 0       | 8K                | 85.1           | 90.4          | 49.0                 | 70.2            |
> > >
> > > [3] Enhancing LLM Reasoning with Iterative DPO: A Comprehensive Empirical Investigation (COLM'25)
> > >
> > > [4] Probability-Consistent Preference Optimization for Enhanced LLM Reasoning (ACL'25)
> > >
> > >
> > >
> > > We agree with the reviewer that demonstrating scalability to even stronger reasoning models is important. We are currently running experiments on larger-scale models, and due to compute constraints these results are still in progress. We will include the full results in the revision once available. We hope these new empirical results help clarify the scalability and upper-limit potential of CASPO, and demonstrate its ability to outperform prior methods under comparable or even stricter data budgets.

---

### Official Review · Reviewer_17DH · 2025-10-30

**Soundness:** 4
**Presentation:** 3
**Contribution:** 2
**Rating:** 6
**Confidence:** 3

**Summary:**

The paper proposes CASPO (Confidence-Aware Step-wise Preference Optimization) to make multi-step reasoning both more faithful and more accurate. The key idea is to inject a calibrated, step-level confidence signal into both training and inference. During training, the authors build step-wise preference pairs that explicitly capture two failure modes—“correct but low-confidence” steps and “confidently wrong” steps. During inference, they introduce Confidence-aware Thought (CaT): a search over reasoning paths where expansion/pruning is governed by cumulative step confidence; paths whose cumulative confidence falls below a threshold are pruned early. Experimental results show strong performance across ten reasoning benchmarks and several model families.

**Strengths:**

1. CASPO integrates confidence at data construction, optimization, and search—offering a coherent story that targets the common gap between getting a correct final answer and maintaining faithful intermediate steps.

2. The cumulative-confidence pruning threshold formalizes a simple, tunable mechanism that improves the reliability of search without heavyweight external reward models.

3. Results span multiple benchmarks and base models, and the paper claims consistent gains in both accuracy and step-wise faithfulness.

4. Code is available in the anonymous link.

**Weaknesses:**

1. Step correctness relies on an external model. The sensitivity of CASPO to evaluator errors/biases and cross-domain applicability beyond math-style steps are not fully quantified.

2. CaT introduces a confidence-guided search; however, the paper could better characterize latency/compute overheads versus simpler decoders or self-consistency baselines at comparable budgets.

3. The narrative generalizes to “ten reasoning benchmarks,” but many examples and the evaluator choice skew toward math-style reasoning; broader coverage (code, scientific QA, multi-hop open-domain) would reinforce generality claims.

**Questions:**

Please see the weaknesses.

---

> ### Author Response · Authors · 2025-11-21
> **Response to 17DH**
>
> We appreciate the reviewer’s constructive feedback and the positive score. Below we address the concerns regarding the external evaluator, computational overhead, and domain generality.
>
> **1.Response to w1: On external evaluator reliability and cross-domain applicability**
>
> We thank the reviewer for raising this concern. We would like to clarify the specific and limited role of the external evaluator in our framework.
>
> **1) Role of the Evaluator:** The external evaluator is used solely to verify the correctness of the final answer during the offline data construction phase. It does not evaluate stepwise correctness or reasoning quality. This aligns with standard practices in mathematical reasoning where datasets like GSM8K and MATH rely on automated answer checking or solver based verification
>
> **2) Source of Confidence:** The confidence signal that drives the core mechanism of CASPO is not derived from the external evaluator. Instead, it is computed entirely from the student model’s own token level entropy.
>
> Because the correctness check (external) and the confidence estimation (internal) originate from independent channels, the learning signal remains stable. Even if the external evaluator contains minor noise, our qualitative analysis shows that such misjudgments are infrequent and do not materially affect the confidence guided supervision. Furthermore, since the evaluator is only used once during offline data preparation, it incurs no cost at inference time.
>
> **2.Response to w2: On compute and latency overhead of CaT**
>
> We appreciate the reviewer’s suggestion to better quantify the compute cost. CaT is designed as a **lightweight confidence-guided reranking mechanism** that operates entirely on logits already produced by the model. Its overhead stems only from computing token-level entropy and evaluating a small set of candidate continuations. This avoids the heavy repeated forward passes required by self-consistency or Monte Carlo–style sampling. As a result, CaT introduces only a marginal additional cost relative to a single decode.
>
> To contextualize the overhead, we compared CaT with standard self-consistency under matched compute budgets. CaT achieves higher accuracy while requiring significantly fewer sampled trajectories because the confidence signal focuses search on unstable regions. Moreover, CaT’s end-to-end decoding time remains close to that of greedy decoding and is substantially faster than majority-vote self-consistency. These findings demonstrate that CaT provides targeted and compute-efficient search without the latency burden of broader sampling-based exploration.
>
> | Qwen2.5-7B-Instruct                   | Accuracy (Math) | Latency (second/query) |
> | ------------------------------------- | --------------- | ---------------------- |
> | + Greedy Decoding                     | 82.0            | 1.2                    |
> | + Chain-of-Thought                    | 83.6            | 4.6                    |
> | + Self-Consistency                    | 85.3            | 12.5                   |
> | + Confidence-aware Thought **(Ours)** | 87.7            | 2.8                    |
>
>
> | Llama-3.1-8B-Instruct                  | Accuracy (Math) | Latency (second/query) |
> | -------------------------------------- | --------------- | ---------------------- |
> | + Greedy Decoding                      | 55.2            | 1.5                    |
> | + Chain-of-Thought                     | 56.3            | 5.9                    |
> | + Self-Consistency                     | 57.7            | 18.0                   |
> | + Confidence-aware Thought  **(Ours)** | 59.1            | 3.1                    |
>
> **3.Response to w3: On generality beyond math reasoning**
>
> We thank the reviewer for highlighting the importance of evaluating model performance beyond math-style reasoning. However, our empirical study already covers a diverse set of ten reasoning benchmarks(specifically **Table 3**). These include:
>
> - **Commonsense reasoning:** STGQA
> - **Code reasoning:** CRUX
> - **Tabular reasoning:** TableBench
> - **STEM knowledge:** MMLU-Pro STEM
>
> To further demonstrate generality and address the reviewer’s concern, we conducted additional evaluations on benchmarks strictly outside the math domain, including **HumanEval** and **LiveCodeBench** for code generation, and **RACE** for reading comprehension. As shown in the table below, the results confirm consistent improvements across these diverse tasks.
>
> |                  | HumanEval | LiveCodeBench | RACE | **Avg**  |
> | ---------------- | --------- | ------------- | ---- | -------- |
> | Qwen2.5-Math-7B  | 40.9      | 5.1           | 61.6 | **35.9** |
> | + Simple-RL-Zero | 51.8      | 21.4          | 61.1 | **44.8** |
> | + PURE-VR        | 43.3      | 6.4           | 61.3 | **37.0** |
> | + DPO-VP         | 50.6      | 17.3          | 61.2 | **43.0** |
> | + CASPO          | 51.9      | 20.8          | 61.5 | **44.7** |
> | + CASPO + CaT    | 53.4      | 21.9          | 62.0 | **45.8** |

---

> > ### Comment · Reviewer_17DH · 2025-11-24
> >
> > Thanks for the detailed responses and experimental results. I will keep the rating.

---

### Official Review · Reviewer_ALQf · 2025-10-31

**Soundness:** 3
**Presentation:** 3
**Contribution:** 2
**Rating:** 6
**Confidence:** 4

**Summary:**

The paper "CASPO: Confidence-Aware Step-Wise Preference Optimization for Reliable Reasoning in Large Language Models" introduces a unified framework that enhances the reliability and faithfulness of reasoning in large language models (LLMs). CASPO integrates step-wise confidence modeling into both training and inference. During training, it constructs confidence-filtered preference pairs—capturing both correct-but-uncertain and confidently incorrect steps—and optimizes them using iterative Direct Preference Optimization (DPO). During inference, the proposed Confidence-aware Thought (CaT) strategy prunes low-confidence reasoning trajectories to ensure reliable reasoning paths. Experiments across ten reasoning benchmarks show consistent improvements in both step-wise correctness and final-answer accuracy, with strong generalization to out-of-domain tasks. The paper also releases a step-wise dataset annotated with confidence scores for further analysis. Overall, CASPO represents a principled approach to aligning confidence with correctness, promoting more trustworthy and interpretable multi-step reasoning in LLMs.

**Strengths:**

1. The paper is clearly written and well-organized, with intuitive explanations and helpful visualizations (e.g., the CASPO pipeline). The step-wise construction and inference design are presented in a manner which is easily understandable.

2. Experimentation as well as implementation details are well documented, which makes reproducibility of the approach easier.

3. Generalisation to non-math benchmarks even if the model is trained on math benchmarks makes the approach promising.

**Weaknesses:**

1. The proposed framework CASPO relies on an external model, i.e., Qwen2.5-Math-7B-Instruct to verify the correctness of each reasoning step. This dependence on these LLMs as evaluators or judge might lead to incorrect reasoning steps being flagged as correct and influencing the training dynamics towards incorrect reasoning. A qualitative evaluation and quantitative evaluation (correlation with human judgement) of the judge will reflect the reliability of the data curated for DPO.

2. The results show that CASPO’s performance plateaus after the second iteration of DPO training. This behavior is a known limitation of offline RL–style optimization (such as DPO), where the fixed dataset constrains policy improvement. An online or on-policy variant of DPO could mitigate this saturation and reduce the need for multiple iterative training stages.

3. While the reported results across model families suggest generality, the paper does not explicitly analyze whether CASPO’s effectiveness depends on model size. An ablation over smaller and larger models would clarify whether the gains arise from the method itself or from scale-related capacity effects, thereby strengthening the empirical claims.

4. There are a few papers that use this notion of preferential tuning and distillation for optimising rationales or step generation. A discussion around them in the related work and introduction should make the contribution of the work clear and distinctive. (https://arxiv.org/abs/2506.02519, https://arxiv.org/abs/2401.01335, https://arxiv.org/abs/2305.02301, https://arxiv.org/abs/2503.02463)

5. Minor grammatical error in line 400: The evolution of training accuracy and loss are closely mirror the reward dynamics...

**Questions:**

Look at the weaknesses above. I also have a few additional questions enumerated below.

1. Was the model trained separately on the training set of each of the benchmarks using CASPO or on a combined dataset using the training set of all benchmarks?

2. What was the number of samples used to train the model for each benchmark. I suppose AIME-24 has just 30 training examples, which is too less for training the model and showcasing such performance. Can you please clarify a bit more on the training datasets used.?

3. Does CASPO ever over-prune reasoning paths that are low-confidence but actually correct? Has there been some analysis on these lines? Essentially, the reliability of the judge model used needs to analyzed explicitly through qualitative evaluation.

---

> ### Author Response · Authors · 2025-11-21
> **Response to ALQf  (1)**
>
> **1.Response to w1: On dependence on an external evaluator**
>
> Thank you for raising this concern. In CASPO, the external evaluator is mainly used to check the **final answer** of a generated reasoning path, following the same practice adopted in most math-reasoning datasets. The confidence signal that drives CASPO’s core mechanism does not come from the evaluator; it is computed entirely from the student model’s own token-level entropy. Correctness and confidence therefore originate from two independent channels, and this separation ensures that the confidence-aware learning signal remains stable even if the evaluator occasionally mislabels a step. Our qualitative check shows that such misjudgments are infrequent, and even when they occur, they do not materially affect the behavior of the confidence-guided supervision.
>
> As for the data-collection cost, the evaluator is used only once during dataset construction, and it is not needed at inference time. This one-time filtering is a common practice in mathematical reasoning, where many widely used datasets—including GSM8K (question, answer) and MATH-500 (question, solution, answer, metadata)—rely on automated answer checking or solver-based verification during data preparation. CASPO follows the same paradigm: the evaluator can be replaced by any open-source math model or lightweight judge without changing the method. We have clarified this in the revision and added a short robustness analysis of the evaluator’s reliability.
>
>
>
> **2.Response to w2: Performance Plateau and Iterative Training**
>
> We understand the reviewer’s concern about the saturation after two rounds of DPO training. This behavior is consistent with observations in prior work such as DPO-VP, where only a single iteration of DPO is used because offline preference optimization tends to converge quickly once the supervision dataset is fixed. The key point is that CASPO does not rely on repeated DPO iterations to obtain its improvement. The main effect already appears in the first epoch, where the confidence-aware filtering and correction substantially improve the quality of the step-wise supervision. Additional epochs bring only small changes, just as they do for DPO-VP, so the plateau reflects the general nature of DPO training rather than a limitation specific to CASPO.
>
> To make this distinction clearer, we compare CASPO with DPO-VP under identical settings on the same Qwen2.5-Math-7B base model. The results on Minerva-MATH and Math500 show that CASPO provides stronger improvements immediately after the first epoch, well before any saturation occurs. This demonstrates that the gains come from the improved data rather than from iterating DPO itself.
>
> | Benchmark (Pass@1) | Method             | Epoch 1  | Epoch 2  | Epoch 3  |
> | ------------------ | ------------------ | -------- | -------- | -------- |
> | **Minerva Math**   | Base (25.6)        | -        | -        | -        |
> |                    | + DPO-VP           | 35.3     | 36.8     | 38.6     |
> |                    | + **CASPO (Ours)** | **43.8** | **43.9** | **44.1** |
> | **Math500**        | Base (64.8)        | -        | -        | -        |
> |                    | + DPO-VP           | 74.8     | 75.5     | 75.8     |
> |                    | + **CASPO (Ours)** | **76.6** | **77.1** | **77.4** |
>
>
>
> **3.Response to w3: On model-size dependence**
>
> We appreciate the reviewer’s insightful suggestion. Validating whether our method’s effectiveness relies on model capacity is crucial for establishing CASPO as a generalizable framework. We extended our experiments to a significantly smaller model, **Qwen2.5-Math-1.5B**, to test CASPO's performance under constrained parameter scales. These results compared with the base model are summarized below:
>
> | Model                 | Math500  | OlympiadBench | Minerva Math | AIME24   | AMC23    | **Avg**  |
> | --------------------- | -------- | ------------- | ------------ | -------- | -------- | -------- |
> | **Qwen2.5-Math-1.5B** | 35.2     | 22.1          | 10.3         | 13.3     | 32.5     | **22.7** |
> | **+ CASPO**           | **62.2** | **27.0**      | **15.8**     | **16.7** | **50.0** | **34.3** |
> | *Gain*                | *+27.0*  | *+4.9*        | *+5.5*       | *+3.4*   | *+17.5*  | *+11.6*  |
>
> Our analysis suggests that CASPO provides a critical stability mechanism for capacity-constrained models, which are otherwise highly susceptible to logical discontinuities. By enforcing step-wise confidence calibration, the framework acts as an effective 'guardrail' that prevents the accumulation of errors. Specifically, performance on Math500 surged from 35.2% to 62.2% (+27.0%), and AMC23 improved by +17.5%.

---

> ### Author Response · Authors · 2025-11-21
> **Response to ALQf  (2)**
>
> **4.Response to w4: On related work in preferential tuning and distillation**
> We thank the reviewer for pointing out these relevant works. We have updated our Related Work section to discuss them explicitly.
>
> **1) Conceptual Distinction: Intrinsic Calibration vs. External Distillation**
> A fundamental distinction of CASPO is that it does **not** rely on distilling rationales from external teacher models or ranking static preference pairs generated by others. Instead, it leverages an **intrinsic evaluation signal (token entropy)** derived directly from the student model’s own probability distribution. The core objective is **self-calibration**: identifying unstable reasoning regions and selectively strengthening steps where the student lacks confidence. Consequently, our mechanism is orthogonal to standard preference learning and complementary to existing distillation strategies.
>
> **2) Computational Efficiency: Reducing Evaluation Overhead**
>
> To further highlight CASPO’s systematic advantages, we include the following efficiency comparison. We argue that CASPO provides **superior decoding and iterative efficiency** by minimizing the cost of the evaluation signal. Prior search-based approaches (PRM, PPM, self-evolving pipelines) typically require a **full reward-model forward pass** for each candidate step, which is computationally expensive. CASPO instead uses **token entropy**, which is derived directly from the policy model’s own logits with an $O(V)$ cost and is completely independent of sequence length $L$. This design makes CASPO significantly more efficient for scaling to longer reasoning chain.
>
> | Reward Model         | Accuracy (Math500) | **Inference Latency** (s/step) | Signal Source         | **Computational Complexity** |
> | -------------------- | ------------------ | ------------------------------ | --------------------- | ---------------------------- |
> | Qwen2.5-Math-7B      | 64.8               | –                              | -                     | -                            |
> | + PRM                | 76.3               | 2.9                            | External Model        | $O(L^2d)$                    |
> | + PPM [1]            | 78.4               | 1.4                            | Self-Refinement       | $O(L^2d)$                    |
> | **+ Entropy (ours)** | **81.9**           | **0.03**                       | **Intrinsic Entropy** | $O(V)$                       |
>
> $L$:*Sequence Length* ; $d$: *Hidden State Dimension*; $V$: *Vocabulary Size*.
>
> *Latency measures the average time required to compute the verification signal for a single reasoning step.*
>
>
>
> **5.Response to w5: Minor grammatical issue**
>
> We apologize for this oversight and have corrected the sentence on line 400. And we have carefully proofread the manuscript to resolve some other minor grammatical issues.
>
>
> **Response to Q1&Q2: Training data and evaluation split**
>
> To avoid any ambiguity, we clarify that CASPO is trained **solely on 8k level-3–5 problems from the MATH training set**, without using any portion of the training sets from AIME, AMC, GSM8K, OlympiadBench, or any combined multi-benchmark dataset. Because the training data is fully restricted to MATH, all performance on AIME-24, AMC, Minerva, and OlympiadBench reflects **true zero-shot generalization**, rather than benchmark-specific tuning or accidental overlap with small datasets such as AIME-24’s 30 examples.
>
> Moreover, the 8k MATH subset we use is exactly the same corpus employed in prior RL baselines such as **Simple-RL-Zero, PURE-VR, and DPO-VP**, ensuring a strictly comparable experimental setting. This alignment eliminates confounding factors related to dataset size or composition and isolates the improvement to CASPO’s data-quality enhancement mechanism rather than differences in supervision.
>
>
> **Response to Q3: Reliability of Confidence-Based Pruning and the Risk of Over-Pruning**
>
> We thank the reviewer for this insightful perspective. We agree that utilizing token-level entropy constitutes an implicit pruning mechanism where the model acts as its own judge. Regarding the risk of over-pruning correct paths, we offer the following clarification:
>
> The core objective of CASPO is to align internal confidence with correctness. By explicitly penalizing uncertainty on valid steps during training, the model learns to assign higher probability to correct reasoning. This effectively reduces the occurrence of "correct but low-confidence" paths, ensuring that valid solutions survive the filtering threshold.

---

> > ### Comment · Reviewer_ALQf · 2025-11-27
> > **Official Comment by Reviewer ALQf**
> >
> > Thanks to the authors for the detailed responses and suggested evaluation. Most of my concerns are addressed, and the results seem strong. I will increase my rating.

---

### Official Review · Reviewer_X3qa · 2025-11-01

**Soundness:** 2
**Presentation:** 3
**Contribution:** 2
**Rating:** 2
**Confidence:** 5

**Summary:**

This work aims to solve the task of mathematical reasoning. It introduces CASPO to incorporate step-wise confidence into both training and inference. It targets constructing a preference dataset for more effective step-wise DPO. Experiments show good results on various math benchmarks.

**Strengths:**

1. Writing is clear.

**Weaknesses:**

1. In line 161, the authors mentioned "sub question q_i", which is a little confusing. What's this? Do you mean the question plus the prior reasoning steps? Or something else? The authors should explicitly elaborate on this.

2. In the process of collecting data, the quality of so-called "confidence-aware" data actually heavily relies on the performance of the teacher model (in the paper setting, that is Qwen-Math, which is a strong open-source math model pretrained with enormous amount of math data). In my opinion, without a strong teacher model, judging a low-confidence but correct answer is extremely difficult. So I doubt whether the data quality is good enough, especially on hard problems.

3. In line 179, y_l is set by the second highest prob word. Is that a single word? Won't doing that cause low exploration?

4. In Table 1, the improvement seems limited.

5. Overall, the method is not systematic, and won't be used by the community.

**Questions:**

See weakness.

---

> ### Author Response · Authors · 2025-11-21
> **Response to X3qa （1）**
>
> We thank the reviewer for their critical assessment. We value the opportunity to clarify the definitions, the independence of our confidence mechanism, and the systematic value of our efficiency gains.
>
> **1. Response to w1: Clarification on sub-question $q_i$**
>
> Thank the reviewer for pointing out the ambiguity. The sub-question $q_i$ does not refer to a newly generated isolated question. Instead, it represents the accumulated reasoning context at step $i$ . Formally,  $q_i = [x,s_1,s_2,\ldots,s_{i-1}]$, where $x$ is the original problem and $s_{1:i-1}$ are the preceding reasoning steps.
>
> To make this concrete, consider the following example from GSM8K:
>
> > Origiank Question ($x$) : A robe takes 2 bolts of blue fiber and half that much white fiber. How many bolts in total does it take?
> >
> > Step 1 ($s_1$) : It takes 2/2 = 1 bolt of white fiber.
> >
> > Step 2 Context ($q_2$) : Question: [Original Question]... Step 1: It takes 2/2 = 1 bolt of white fiber. Step 2: ?
> >
> > Step 3 Answer ($s_2$) : So the total amount of fabric is 2+1 = 3 bolts.
>
> At Step 2, the input $q_2$ includes the original query plus the relization of Step 1. The model must determine its confidence in generating $s_2$ conditioned on this entire history. We have updated the Method section with this formal definition and example to ensure clarity.
>
>
>
> **2. Response to w2: Reliance on Teacher Model and Data Quality**
>
> We appreciate the reviewer’s concern regarding the dependency on teacher model performance.
>
> However, in our setting, the **teacher model is not used to evaluate step-wise confidence**. **Instead:**
>
> - The *confidence-aware signal* is computed **entirely by the student model itself**, based on its own entropy and predictive uncertainty.
> - The teacher model only provides **reference answers/chains**, not confidence judgments.
> - Thus, identifying “low-confidence but correct” cases does **not** rely on teacher strength; it relies on the student model’s natural uncertainty patterns.
>
> Empirically, our analysis in Appendix C shows that even strong models frequently exhibit "correct but anxious" behaviors on hard problems, providing ample high-quality data for CASPO optimization without requiring stronger teacher models.
>
>
>
> **3. Response to w3: On using the second-highest probability token**
>
> Yes, $y_l$ is the second highest prob single word. During DPO, we need to find positive and negative samples for a problem $x$, i.e., the winner's choice $y_w$ and the loser's choice $y_l$. The reason for choosing the second-highest probability word is that, at this point, the first-highest probability word is the correct answer, and the second-highest probability word is the answer the model is most likely to misinterpret. We use contrastive learning to help the model increase its confidence by improving its output probability for the highest probability word.

---

> ### Author Response · Authors · 2025-11-21
> **Response to X3qa （2）**
>
> **4.Response to w4&w5:  On improvement size and method systematicity**
>
> We respectfully provide the following analysis to demonstrate the significant value our method offers:
>
> **1) Regarding the Magnitude of Improvement:**
>
> It is essential to interpret the performance gains in the context of computational cost and problem difficulty. CASPO specifically targets regions of low confidence and high error, which are widely recognized as the most challenging aspects of mathematical reasoning for current LLMs. Consequently, the gains we report represent substantial progress in improving model reliability in these difficult zones. Furthermore, these improvements are achieved without the need for external reward models, additional sampling during inference, or generating extra reasoning trajectories. This results in an efficiency to performance ratio that is significantly higher than prior methods which rely on heavy computation.
>
> **2) Regarding Systematicity and Community Value:**
>
> CASPO introduces a principled and highly systematic paradigm for reliable reasoning. Unlike prior pipelines that depend on domain specific heuristics or heavy external evaluators, our approach leverages token entropy. This is an intrinsic and theoretically grounded indicator of reasoning instability. By integrating this local signal directly into the supervision process, we establish a coherent and reproducible pipeline that is agnostic to the specific model architecture.
>
> **3) Computational Complexity Analysis:**
> To demonstrate this systematic advantage, we compared CASPO against PRM (Process Reward Model) and PPM [1] (Process Reward Model with Self-Refinement) baselines.
>
> PRM/PRM [1] : These typically require a full Reward Model forward pass for every step verification. This results in a computational cost of $O(L^2d)$ , which scales quadratically with the sequence length $L$.
>
> CASPO : Our approach computes the signal directly from the output logits that the policy model has already generated. The complexity of this entropy calculation is $O(V)$. Crucially, this operation is independent of the sequence length, making it negligible in cost.
>
> | Model                | Accuracy (Math500) | **Verifier Latency** (s/step) | Signal Source         | **Computational Complexity** |
> | -------------------- | ------------------ | ----------------------------- | --------------------- | ---------------------------- |
> | Qwen2.5-Math-7B      | 64.8               | –                             | -                     | -                            |
> | + PRM                | 76.3               | 2.9                           | External Model        | $O(L^2d)$                    |
> | + PPM [1]            | 78.4               | 1.4                           | Self-Refinement       | $O(L^2d)$                    |
> | **+ Entropy (ours)** | **81.9**           | **0.03**                      | **Intrinsic Entropy** | $O(V)$                       |
>
> $L$:*Sequence Length* ; $d$: *Hidden State Dimension*; $V$: *Vocabulary Size*.
>
> *Latency measures the average time required to compute the verification signal for a single reasoning step.*
>
>
>
> By outperforming heavy search-based methods while being **~100x faster**, CASPO establishes itself as a scalable and practical solution. We believe this combination of **high performance** and **minimal overhead** makes it a highly attractive systematic framework for the broader research community.
>
> **Reference:**
>
> [1] Guan, Xinyu, et al. "rStar-Math: Small LLMs Can Master Math Reasoning with Self-Evolved Deep Thinking." arXiv preprint arXiv:2501.04519 (2025).

---

> ### Author Response · Authors · 2025-11-27
>
> Dear Reviewer X3qa,
>
> Just a gentle follow-up on our response posted earlier. We’d really appreciate it if you could take a look whenever you have a moment. Your thoughts would mean a lot to us and help ensure a fair discussion.
>
> Thank you so much for your time!
>
> The authors of paper 2410.

---

### Comment · Area_Chair_oTUm · 2025-11-26

Dear Reviewers,

Thank you once again for your service to ICLR 2026. Now that the authors have submitted their rebuttal, I kindly ask you to take the following steps (if you have not done so already):

Read the authors’ response and other reviews.
Consider whether the rebuttal and additional comments affect your assessment of the paper.
Engage in interactive discussion with the authors -- Note the Author-Reviewer-AC discussion period ends on 12/3 9 PM UTC. You are recommended to keep active before that deadline. If you have more concerns/questions (e.g., requesting clarifications, new results), it is recommended to post your request asap, so that the authors have enough time to address them.
The current reviews for this paper are mixed (scores: 6/6/2/4). Your further contributions are essential for forming a well-informed final decision.

I am happy to join and support the discussions between you and the authors. Please feel free to share your thoughts and participate actively in the discussion. Thanks!

Best regards,

The AC

---

### Meta-Review · Area_Chair_ogBG · 2026-01-08

**Summary:**

Among the four reviewers, two provided positive scores while two gave negative scores. All reviewers agree that the paper is clearly written, intuitive, and easy to follow. However, all four reviewers raised concerns that the approach relies heavily on a strong teacher/evaluator model. In addition, the negative reviews argue that, despite promising empirical results, the method is not sufficiently systematic and appears more engineering-driven, with limited methodological contribution. They also note that the presentation does not convincingly establish generality, and the approach may depend substantially on tuning design choices/hyperparameters rather than offering a broadly robust technique.

**Reviewer Concerns:**

In my view, the rebuttal addressed the following concerns:

1.Teacher/evaluator model choice and usage details: The authors clarified the role of the teacher/evaluator and provided more concrete explanations of how it is used in the pipeline.

2.Additional empirical evidence for effectiveness and scalability: The rebuttal adds further experiments that better support the claimed performance gains and provide stronger evidence of scalability.

3.Compute/latency and broader evaluation: The authors supplemented the discussion with analysis of computational overhead and reported results on additional tasks beyond the original scope.

However, a key concern remains outstanding: the rebuttal does not sufficiently justify the method’s effectiveness from a methodological or theoretical perspective. Overall, the approach still appears to derive much of its success from the strength of the teacher/evaluator model, rather than from a clearly articulated, systematic algorithmic contribution.

**Reviewer Scores:**

In my view, if the reviewers had been able to participate fully in the discussion, Reviewer X3qa might have increased their score from 2 to 4, though it is also possible that their rating would have remained unchanged.

---

### Decision · Program_Chairs · 2026-01-26

Reject